# Systematic Review and Network Meta-Analysis of Immune Checkpoint Inhibitors in Combination with Chemotherapy as a First-Line Therapy for Extensive-Stage Small Cell Carcinoma

**DOI:** 10.3390/cancers12123629

**Published:** 2020-12-03

**Authors:** Hsiao-Ling Chen, Yu-Kang Tu, Hsiu-Mei Chang, Tai-Huang Lee, Kuan-Li Wu, Yu-Chen Tsai, Mei-Hsuan Lee, Chih-Jen Yang, Jen-Yu Hung, Inn-Wen Chong

**Affiliations:** 1Department of Pharmacy, Kaohsiung Municipal Ta-Tung Hospital, Kaohsiung 80145, Taiwan; hlchen369@gmail.com (H.-L.C.); 880504@kmhk.org.tw (H.-M.C.); 2Institute of Epidemiology and Preventive Medicine, National Taiwan University, Taipei 100225, Taiwan; yukangtu@ntu.edu.tw; 3Department of Medical Research, National Taiwan University Hospital, Taipei 100225, Taiwan; 4Department of Internal Medicine Kaohsiung Municipal Ta-Tung Hospital, Kaohsiung Medical University, Kaohsiung 88145, Taiwan; weatlee@gmail.com (T.-H.L.); jenyuhung@gmail.com (J.-Y.H.); 5Division of Pulmonary and Critical Care Medicine, Department of Internal Medicine, Kaohsiung Medical University Hospital, Kaohsiung Medical University, Kaohsiung 88708, Taiwan; 980448kmuh@gmail.com (K.-L.W.); 1010362kmuh@gmail.com (Y.-C.T.); 6Department of General Medicine, Kaohsiung Medical University Hospital, Kaohsiung Medical University, Kaohsiung 88708, Taiwan; chong@kmu.edu.tw; 7Faculty of Medicine, College of Medicine, Kaohsiung Medical University, Kaohsiung 88708, Taiwan; 8Respiratory Therapy, College of Medicine, Kaohsiung Medical University, Kaohsiung 88708, Taiwan; 9Department of Biological Science & Technology, National Chiao Tung University, Hsinchu 300, Taiwan

**Keywords:** immune checkpoint inhibitors, extensive-stage small cell lung cancer, nivolumab, ipilimumab, pembrolizumab, atezolizumab, durvalumab, chemotherapy

## Abstract

**Simple Summary:**

Patients with extensive-stage small cell lung cancer (ED-SCLC) have a very short survival time even if they receive standard chemotherapy. Currently, the combination of chemotherapy plus immune checkpoint inhibitors (ICIs) as the first line treatment had superior survival than chemotherapy alone in randomized control trials. However, there is a lack of head-to-head comparisons for these combination regimens. We conducted a systematic review and network meta-analysis to provide a treatment ranking of ICIs for ED-SCLC. In summary, the probability of nivolumab was associated with the best ranking for overall survival, followed by atezolizumab, durvalumab, pembrolizumab, and ipilimumab. The ranking of progression free survival from the best to the worst was as follows: nivolumab, pembrolizumab, atezolizumab, durvalumab, and ipilimumab. However, nivolumab had the highest probability of grade 3–4 adverse events in our study. Further head-to head large-scale phase III randomized controlled studies are needed to verify our conclusions.

**Abstract:**

Patients with extensive-stage small cell lung cancer (ED-SCLC) have a very short survival time even if they receive standard cytotoxic chemotherapy with etoposide and platinum (EP). Several randomized controlled trials have shown that patients with ED-SCLC who received a combination of EP plus immune checkpoint inhibitors (ICIs) had superior survival compared with those who received EP alone. We conducted a systematic review and network meta-analysis to provide a ranking of ICIs for our primary endpoints in terms of overall survival (OS), progression free survival (PFS), and objective response rate (ORR), as well as our secondary endpoint in terms of adverse events. The fractional polynomial model was used to evaluate the adjusted hazard ratios for the survival indicators (OS and PFS). Treatment rank was estimated using the surface under the cumulative ranking curve (SUCRA), as well as the probability of being best (Prbest) reference. EP plus nivolumab, atezolizumab or durvalumab had significant benefits compared with EP alone in terms of OS (Hazard Ratio HR = 0.67, 95% Confidence Interval CI = 0.46–0.98 for nivolumab, HR = 0.70, 95% CI = 0.54–0.91 for atezolizumab, HR = 0.73, 95% CI = 0.59–0.90 for durvalumab) but no significant differences were observed for pembrolizumab or ipilimumab. The probability of nivolumab being ranked first among all treatment arms was highest (SCURA = 78.7%, Prbest = 46.7%). All EP plus ICI combinations had a longer PFS compared with EP alone (HR = 0.65, 95% CI = 0.46–0.92 for nivolumab, HR = 0.77, 95% CI = 0.61–0.96 for atezolizumab, HR = 0.78, 95% CI = 0.65–0.94 for durvalumab, HR = 0.75, 95% CI = 0.61–0.92 for pembrolizumab), and nivolumab was ranked first in terms of PFS (SCURA = 85.0%, Prbest = 66.8%). In addition, nivolumab had the highest probability of grade 3–4 adverse events (SUCRA = 84.8%) in our study. We found that nivolumab had the best PFS and OS in all combinations of ICIs and EP, but nivolumab also had the highest probability of grade 3–4 adverse events in our network meta-analysis. Further head-to head large-scale phase III randomized controlled studies are needed to verify our conclusions.

## 1. Introduction 

Small cell lung cancer (SCLC) accounts for 10–15% of all lung cancer cases and is known for its aggressive behavior, rapid doubling time, growth, and early spread to distant sites. The most common risk factor for SCLC is smoking tobacco, and up to 98% of patients with SCLC have a history of smoking. SCLC is characterized by multiple genetic alterations, reflecting its genomic instability [1,2]. The majority of SCLCs express alterations in chromosome 3p and mutations in *RB1, TP53, RASSF1*, *MYC*, *FGFR1*, and *PTEN* [3,4]. In addition to these genomic alterations, there are also malfunctions in specific regulatory pathways. Long term exposure to tobacco smoke causes an increase in the tumor mutation burden (TMB) and SCLC is associated with a higher expression of DNA damage response (DDR) pathway mediators [5]. Early detection of SCLC is very challenging due to the lack of specific symptoms and the rapid tumor growth, making current approaches to screening ineffective for diagnosing patients at early disease stages [6,7,8]. Around 70% of cases present with extensive-stage SCLC at diagnosis (ED-SCLC); the remaining 30% of patients have limited-stage SCLC (LD-SCLC) [8]. First-line standard chemotherapy is a combination of etoposide with platinum (EP) [6,7]. In ED-SCLC, chemotherapy is the mainstay treatment in the first-line setting. The median overall survival (OS) rates range from 15 to 20 months for LS-SCLC and 8 to 13 months for ED-SCLC. The five-year survival rate is 20% to 25% for LS-SCLC, but only about 2% for ED-SCLC, and there is an average OS period of only two to four months for untreated ED-SCLC patients [6]. SCLC is usually sensitive to the initial chemotherapy treatment; however, most patients develop recurrent disease, often with metastasis to additional sites after the initial treatment. Currently, radiation therapy to the chest and prophylactic cranial irradiation are applied to destroy undetectable cancer cells and decrease the risk of recurrence. Topotecan is a standard second-line treatment choice but its efficacy is very limited [6]. There is no standard of care beyond second-line therapy. Systemic therapy for SCLC patients has not changed substantially in several decades [2,6]. Consequently, there is an urgent medical need to bring new treatment options to SCLC patients.

SCLC is a tumor with one of the highest rates of somatic mutations and this characteristic can result in a higher likelihood of identifying tumor-specific neoantigens that may ultimately trigger an adaptive immune response that is capable of detecting and eradicating tumor cells [1,2]. Preclinical data has demonstrated that treatment with antibodies specific for anti-cytotoxic T lymphocyte associated antigen 4 (CTLA-4) can restore an immune response through the increased accumulation and survival of memory T cells and depletion of regulatory T cells (Tregs). The use of monoclonal antibodies (mAbs) to block either programmed cell death 1 (PD-1) or anti-programmed cell death ligand 1 (PD-L1) prevents the downregulation of T cell effector function, allowing T cells to mediate tumor cell death. Several phase III trials reported that immune checkpoint inhibitors (ICIs) in combination with chemotherapy had clinical benefit in terms of progression free survival (PFS) and OS when used as a salvage therapy for advanced non-small cell lung cancer compared with those patients who received chemotherapy alone [9,10,11,12]. Since then, several trials have been designed to combine chemotherapy with ICIs, including anti-CTLA4, anti-PD-1 or anti-PD-L1, as a first line therapy for ED-SCLC [13,14,15,16,17,18]. Except for anti-CTLA4, ICIs plus chemotherapy provided a better survival benefit for newly diagnosed ED-SCLC. However, there is a lack of head-to-head comparisons for these combination regimens in the treatment of ED-SCLC. As ICIs are very expensive and ED-SCLC patients have a short survival time, we urgently need to know which ICI plus EP is most effective and reliable. Therefore, we used PFS, OS, objective response rate (ORR) and grade 3–4 adverse drug reactions as the major outcomes in a network meta-analysis of current randomized phase II and phase III trials which reported on ICI plus EP treatment to evaluate their clinical efficacy.

## 2. Results

### 2.1. Literature Search

A total of 393 studies were identified following electronic searches, and eight studies were identified from American Society of Clinical Oncology (ASCO) and European Medical Oncology (ESMO). After the exclusion of duplicate studies, 211 papers underwent title/abstract screening. Of those, 82 were excluded due to being incomplete randomized controlled trials (RCTs)and 93 were excluded on the basis of the title/abstract review, leaving 36 studies that underwent a full text review. At the end of the review process, six met the inclusion criteria and underwent qualitative synthesis and quantitative meta-analysis. A Preferred Reporting Items for Systematic Reviews and Meta-Analyses (PRISMA) flow diagram is presented in Figure 1 and the reasons for exclusion are provided in Appendix A.

#### Study Characteristics and Quality Evaluation

The characteristics of the included studies are provided in Table 1. All six trials were phase II or III and were completed between 2013 and 2020 for newly diagnosed ED-SCLC patients who had not received previous treatment. Except for CA184-041 [16], the control group of all included trials was chemotherapy with etoposide plus platinum (EP) agents. There were two study arms in four of the trials which compared chemotherapy plus ICIs, such as ipilimumab [17], nivolumab [14], pembrolizumab [18], or atezolizumab [13], to chemotherapy alone, while there were three study arms in the CASPIAN [15] and CA184-041 trials. However, the groups with durvalumab, tremelimumab and EP in the CASPIAN trial did not meet the predefined statistical significance threshold at the time of the interim analysis and therefore the result was not presented in the final report. Besides, the group with the concurrent regimen in CA184-041 [16], a phase II trial for ipilimumab, was not regarded as a treatment arm in CA184-156 [17], a phase III trial for ipilimumab, due to there being no improvement in efficacy. The percentage of patients aged ≥ 65 years across the trials ranged from 23.1% to 52.3%, and the percentage of males ranged from 44.4% to 78.4%. The percentage of brain or CNS metastases ranged from 8.68% to 12.14% across all included trials. Response evaluation criteria in solid tumors version 1.1 (RECIST) was used to assess tumor shrinkage (objective response) and disease progression in four of the trials. The other two trials for ipilimumab (CA184-041 and CA184-156) assessed the tumor burden using the modified World Health Organization (mWHO) criteria, as well as the immune-related (IR) response criteria. In conclusion, CA-184-041 [16] was excluded from our analysis due to heterogeneity in the chemotherapy regimen. Furthermore, data from CA184-156 was not included in the network comparisons for PFS and ORR because different criteria were used for cancer progression and treatment response.

The results of the quality assessment are presented in Appendix A. Detailed information about EA5161 was determined from the protocol in ClinicalTrials.gov because limited data was provided in the report from the 2020 annual meeting of the American Society of Clinical Oncology. Both CASPIAN and EA5161 were open label studies; therefore, a high risk of bias was declared for blinding. Besides, unclear assessments were presented and resulted from a lack of detailed information about the random sequence generation and allocation process.

### 2.2. Pooled Results for ICIs and Their Effect on Efficacy and Safety

Pooled results for the effect of different ICIs on OS, PFS and ORR as well as grade 3–4 adverse events are provided in Appendix A. Compared to chemotherapy alone, ICI plus chemotherapy significantly increased the OS and PFS but no significant effect was observed for ORR. On the other hand, ICI plus chemotherapy slightly increased the risk of grade 3–4 adverse events. With regards to the blockade of ICIs, anti-PD-1 agents (nivolumab and pembrolizumab) were associated with significant benefits in OS and PFS. A noticeable OS benefit was seen in the patients who received anti-PD-L1 agents (atezolizumab and durvalumab), but there were no significant effects in PFS and ORR. Only one RCT evaluated the efficacy of an anti-CTLA4 agent (ipilimumab), but there was no improvement in OS between the patients who received ipilimumab plus chemotherapy and those who received chemotherapy alone. In terms of safety, no statistical risks were reported among the patients who received ICIs plus chemotherapy, regardless of the subgroup of ICIs.

#### Efficacy and Safety Evaluation from the Network Meta-Analysis

The network constructions are presented in Figure 2. For OS and grade 3–4 adverse events, five ICIs plus chemotherapy and chemotherapy alone were included in the network meta-analysis. In terms of PFS and ORR, four ICIs plus chemotherapy and chemotherapy alone were included in the network meta-analysis. The effect sizes of the pairwise comparisons are summarized in Figure 3 and the surface under cumulative ranking curve (SUCRA) rankings are detailed in Figure 4. The probability of being the best treatment is shown in Figure 5 for all efficacy and safety indicators.

### 2.3. Efficacy and Safety Evaluation

In terms of pairwise comparisons for OS (Figure 3A), chemotherapy plus nivolumab, atezolizumab or durvalumab gave a significantly improved benefit compared with chemotherapy alone (HR = 0.67, 95% CI = 0.46–0.98 for nivolumab, HR = 0.70, 95% CI = 0.54–0.91 for atezolizumab, HR = 0.73, 95% CI = 0.59–0.90 for durvalumab). However, the efficacy was shown to be similar between pembrolizumab or ipilimumab plus chemotherapy and chemotherapy alone (HR = 0.80, 95% CI = 0.64–1.00 for pembrolizumab, HR = 0.92, 95% CI = 0.80–1.09 for ipilimumab). Although no superior effects were indicated in pairwise comparisons between different ICIs, anti-PD-1 agents (pembrolizumab or nivolumab) and anti-PD-L1 agents (atezolizumab or durvalumab) had lower HRs compared with the anti-CTLA4 agent (ipilimumab). Regarding the treatment efficacy ranking, the probability showed that nivolumab was associated with the best ranking for OS (highest SCURA and Prbest value; SCURA = 78.6%, Prbest = 46.7%), followed by atezolizumab (SCURA = 75.7%), durvalumab (SCURA = 68.9%), pembrolizumab (SCURA = 51.3%), ipilimumab (SCURA = 20.4%) and chemotherapy alone (SCURA = 5.0%). Based on the results from subgroup analysis (Appendix A), ranking was similar to the overall subjects. Although durvalumab was not regarded as being a better ICI in OS, it was recommended as the best choice for females younger than 65 years old with brain metastasis at baseline.

In terms of PFS (Figure 3B), all ICIs included in our analysis improved PFS compared with chemotherapy alone (HR = 0.65, 95% CI = 0.46–0.92 for nivolumab, HR = 0.77, 95% CI = 0.61–0.96 for atezolizumab, HR = 0.78, 95% CI = 0.65–0.94 for durvalumab, HR = 0.75, 95% CI = 0.61–0.92 for pembrolizumab). Even though the different ICIs had a similar effect on PFS, treatment with nivolumab had a lower HR than the others. Additionally, the probability showed that nivolumab was associated with the best ranking for PFS (highest SUCRA and Prbest; SCURA = 85.0%, Prbest = 66.8%). As for the blockade of ICIs, anti-PD-1 agents (SCURA = 85.0% for nivolumab and 60.8% for pembrolizumab) were associated with better rankings than anti-PD-L1 agents (SCURA = 54.2% for atezolizumab and 49.5% for durvalumab). Chemotherapy alone had the lowest score (SCURA = 0.6%).

In terms of the objective response rate (Figure 3C), durvalumab was associated with a superior ranking compared with chemotherapy alone (response ratio = 1.18, 95% CI = 1.03–1.34) but no significant differences were observed between the other ICIs and chemotherapy alone. Based on pairwise comparisons between ICIs, no significant difference was observed between any comparable ICIs except for durvalumab, which produced a noticeable benefit over atezolizumab. Moreover, durvalumab was regarded to have a better objective response rate. Durvalumab had the highest SUCRA and Prbest scores (SCURA = 82.1%, Prbest = 46.2%), followed by pembrolizumab (SCURA = 71.8%) and nivolumab (SCURA = 57.4%). However, adding atezolizumab to chemotherapy treatment did not give a better ranking compared with chemotherapy alone, as the SCURA value of atezolizumab was the lowest (SUCRA = 28.3% for chemotherapy alone and SCURA = 10.4% for atezolizumab).

In terms of pairwise comparisons for grade 3–4 adverse events (Figure 3D), we found no significant differences in the risk of grade 3–4 adverse events between any two treatment arms. Although a higher risk was observed among nivolumab users, the risk ratios were close to one and there were no statistical differences. Based on the SUCRA value, a larger SUCRA value indicated a higher treatment risk. Nivolumab (SUCRA = 86.9%, Prbest = 61.3%) had the highest probability of grade 3–4 adverse events, followed by ipilimumab (SUCRA = 51.3%), pembrolizumab (SUCRA = 49.4%), atezolizumab (SUCRA = 42.3%), chemotherapy alone (SUCRA = 37.4%) and durvalumab (SUCRA = 32.7%). Finally, death events for the included RCTs are summarized in Appendix A. Due to the limited data that was provided in the published RCTs, it was not possible to classify death events into any events leading to death, immune-mediated adverse events leading to death, or chemotherapy leading to death, so we did not conduct network meta-analyses for death events. The ranking of specific toxicities was similar to the ranking of overall adverse events, except for ipilimumab (Appendix A). Ipilimumab had a lower risk for anemia (ranking = five of six), neutropenia (ranking = six of six) and thrombocytopenia (ranking = six of six); however, ipilimumab had worse safety data for overall adverse events (ranking = two of six).

## 3. Discussion

ICI plus chemotherapy has been shown to be superior to traditional chemotherapy in both OS and PFS for patients with treatment naïve ED-SCLC in several trials [13,14,15,18]. Our network meta-analysis firstly proposed a ranking for PFS, OS, ORR and grade 3–4 adverse events for different combinations of ICI and chemotherapy for ED-SCLC treatment.

Add-on ICIs are a feasible way of improving the very short survival time of ED-SCLC patients who receive the standard EP regimen [13,14,15,18]. Although a response rate of 50% to 80% is achieved in first-line treatment with EP chemotherapy, progression occurs rapidly and there is only a 15–20% response to secondary topotecan chemotherapy [6,8]. Since SCLC has a very short survival time, new treatment strategies are urgently needed. The clinical efficacy of immunotherapies has been observed in patients with refractory or metastatic SCLC [12,19,20,21,22,23]. The phase II KEYNOTE-158 study [21] showed that the PFS of pembrolizumab for relapsed SCLC was 2.0 months, the median OS was 9.1 months, and the one-year PFS and OS were 16.8% and 40.2%, respectively. In a non-randomized cohort of patients with advanced SCLC treated in the CheckMate 032 study [19], the estimated two-year OS rate was 14% with nivolumab monotherapy and 26% with nivolumab plus ipilimumab. Nivolumab plus ipilimumab appeared to provide a greater clinical benefit than nivolumab monotherapy in SCLC patients with a high tumor mutation burden [5]. Furthermore, several trials were designed as a first line therapy for ED-SCLC.

A randomized phase II study (CA184-041) led by Reck et al. was designed to compare paclitaxel with carboplatin plus ipilimumab and paclitaxel with carboplatin alone, but the results did not reveal a significant difference in the ORR, PFS or OS between the treatments [16]. A randomized phase III trial (CA184-156) was further designed to compared ipilimumab and EP with EP alone, and the study showed a minimal increase in median PFS, but there was no significant change in median OS [17]. Recently, several phase II and III trials have been designed to evaluate the addition of PD-1 and PD-L1 inhibitors to EP, and they have demonstrated positive results [13,14,15,18]. Some traditional meta-analyses of the efficacy of ICIs plus EP have shown that a combination of EP plus ICIs gives superior PFS and OS compared with EP alone for the treatment of ED-SCLC [24,25]. However, the current network meta-analysis proposed not only the efficacies, but also showed the ranking of these ICIs plus chemotherapy combinations on PFS, OS, ORR and grade 3–4 adverse events in different combinations. Firstly, our study showed that all ICIs with EP combination regimens had superior PFS compared with EP treatment alone. All combinations of ICIs with EP enrolled in our network meta-analysis improved the PFS compared with chemotherapy alone. Among all ICI plus EP combinations, nivolumab plus EP had the lowest HR, and nivolumab plus EP ranked first in the treatment of ED-SCLC for PFS and could be regarded as the most reliable combination among all evaluated regimens. In addition, anti-PD-1 agents plus EP ranked better than anti-PD-L1 agents plus EP and EP alone in our analysis.

EP plus nivolumab, atezolizumab or durvalumab all had a significantly better ranking for OS compared with EP alone, but no significant benefits were observed for pembrolizumab or ipilimumab plus EP. Anti-PD-1 agents and anti-PD-L1 agents had lower HRs compared with the anti-CTLA4 agent. In fact, ipilimumab plus paclitaxel and carboplatin failed to demonstrate the efficacy of paclitaxel and carboplatin alone in a phase II trial and it also failed in a phase III study when compared with ipilimumab plus EP [16,17]. Furthermore, nivolumab was ranked as the most optimal ICI among all ICIs plus EP in the treatment arms, followed by atezolizumab, durvalumab, pembrolizumab, ipilimumab and EP alone.

Some previous trial results indicated that ICIs plus chemotherapy were better than chemotherapy alone in terms of ORR and disease control rate (DCR) [15,18]. However, some showed that there was no significant difference in the ORR and DCR between ICIs plus chemotherapy and chemotherapy alone [13,14]. In our analysis, ICIs with EP were compared with EP alone in terms of ORR and DCR. Durvalumab was determined to be the optimal treatment regimen with a better objective response rate, but no significant differences were presented between other ICIs and chemotherapy alone in our network meta-analysis. Interestingly, there was a discrepancy between the ranking of ORR benefit and OS/PFS benefit. This discrepancy may have been caused by measurement bias due to how tumor measurements were taken (in the setting of the subjectivity of RECIST) and also when these measurements were made [26].

Clinically, grade 3–4 adverse events always limit the application of effective combinations of ICIs and EP and grade 3–4 adverse events become the main concern [9,16]. Some previous analyses of these trials indicated that there were fewer serious hematology-related toxicities for the ICI plus chemotherapy group compared with the chemotherapy alone group, however, serious non-hematology-related toxicities were more common in patients receiving an ICI combined with chemotherapy, and there were significant increases in fatigue, rashes, diarrhea, and elevated aminotransferase enzymes [16,27]. In the current study, we found non-significant differences in the risk of grade 3–4 adverse events between any two treatment arms, but nivolumab plus EP had the highest risk. Among all ICIs, nivolumab plus EP had the highest probability of grade 3–4 adverse events as determined by the SUCRA ranking method.

Our meta-analysis had some limitations. First, heterogeneity was present among these included RCTs, such as the regimen for chemotherapy and criteria for treatment response or progression. Therefore, the CA184-041 phase II RCT was excluded from the quantitative synthesis. In addition, the EA5161 study is a small sample phase II trial assessing the effect of nivolumab on ED-SCLC. Although narrow Cls provided strong evidence in EA5161, further RCTs with large sample sizes are needed to confirm our findings. Therefore, our results only serve as a platform for future trials that attempt to introduce nivolumab as a first-line therapy for ES-SCLC in a large phase III RCT, not as direct evidence to promote nivolumab-based therapy as a frontline option at present.

## 4. Methods

This study was conducted in accordance with the Preferred Reporting Items for Systematic Reviews and Meta-Analyses (PRISMA) guidelines. A prospective protocol was created in advance and registered on the International Prospective Register of Systematic Reviews PROSPERO website (registration number: CRD42020215762) [27].

### 4.1. Search Strategy and Study Selection

A comprehensive literature search was performed of PubMed, Embase, Cochrane library ClinicalTrials.gov, the database of the American Society of Clinical Oncology (ASCO) and the dataset of European Medical Oncology (ESMO) from their conception until 30 September 2020 without a language limitation. Full details of the search strategy are presented in Appendix A and the search keywords were as follows: small cell lung cancer (SCLC or small cell lung carcinoma), immune checkpoint inhibitors (ICIs) anti-CTLA4, anti-PD-1 or anti-PD-L1, and the specific names of drugs (ipilimumab, nivolumab, durvalumab, pembrolizumab, atezolizumab, lambrolizumab, avelumab or tremelimumab). In order to include more relevant studies, controlled vocabulary search terms for PubMed (MeSH) and Embase (Emtree) were used and additional references were sought from the reference lists of the retrieved studies. The inclusion criteria were as follows: (1) completed phase II–III randomized control trials (RCTs) involving adults with ED-SCLC; (2) the RCTs involved newly diagnosed untreated patients with ED-SCLC; (3) the RCTs investigated and compared the efficacy and safety of an ICI combined with chemotherapy with chemotherapy alone.

### 4.2. Data Extraction and Quality Assessment

Two independent reviewers (H.L. Chen and C.J. Yang) performed the data extraction and quality assessment. Any unresolved discrepancies in the data extraction or appraisal of the results were resolved by discussion with a third reviewer (M.S. Lee). The extracted information included trial details, such as trial name, year published, phase of trial, baseline characteristics, regimen and patient number, primary endpoints, secondary endpoints, and criteria for treatment response. Only grade 3–4 adverse events as defined by the Common Terminology Criteria for Adverse Events (CTCAE) were included for the safety analysis [28]. Quality assessment was performed using the Cochrane Collaboration’s Risk of Bias tool. Quality assessment was performed using Review Manager version 5.1 [29].

### 4.3. Data Synthesis and Statistical Analysis

Treatment efficacies were evaluated in terms of OS, PFS and ORR. The safety outcomes focused on grade 3–4 adverse events as determined by the CTCAE. A fractional polynomial model was used to evaluate the adjusted hazard ratios (HRs) for the survival indicators (OS and PFS). In terms of binomial indicators, response ratio was regarded as the effect size for the objective response rate and risk ratio was used for adverse events along with 95% confidence intervals (CIs).

We first generated the network graphs for different outcomes separately to determine which treatments were directly or indirectly comparable. After that, we performed a frequentist network meta-analysis to estimate the comparative effect of each pair of treatments included in the constructed network. Fixed-effect models were used, since in most cases the treatment of interest was evaluated in one trial and the number of included trials per comparison was too small to estimate between-study heterogeneity. Finally, treatment rank was estimated by a surface under cumulative ranking curve (SUCRA), as well as the probability of being best (Prbest). SUCRA was computed for a more precise estimation of the ranking probabilities and the larger the SUCRA value, the better the intervention. The Prbest value indicated that the treatment was the best choice in the top rank. All statistical analyses and network graph generation were performed using Stata 11.2 [30].

## 5. Conclusions

In summary, our network meta-analysis showed that EP plus all ICIs have longer PFS compared with chemotherapy alone, while nivolumab ranked first in the SUCRA ranking analysis. Furthermore, EP plus nivolumab, atezolizumab or durvalumab all provided significant improvements for OS compared with EP alone, and nivolumab also ranked first among all treatment arms. In addition, durvalumab plus EP showed a better objective response rate than the other ICIs plus EP. Finally, nivolumab had the highest probability of grade 3–4 adverse events according to the SUCRA ranking. Given the limited number of studies included in the meta-analysis, additional large-scale phase III RCT studies are needed to verify these conclusions.

## Figures and Tables

**Figure 1 cancers-12-03629-f001:**
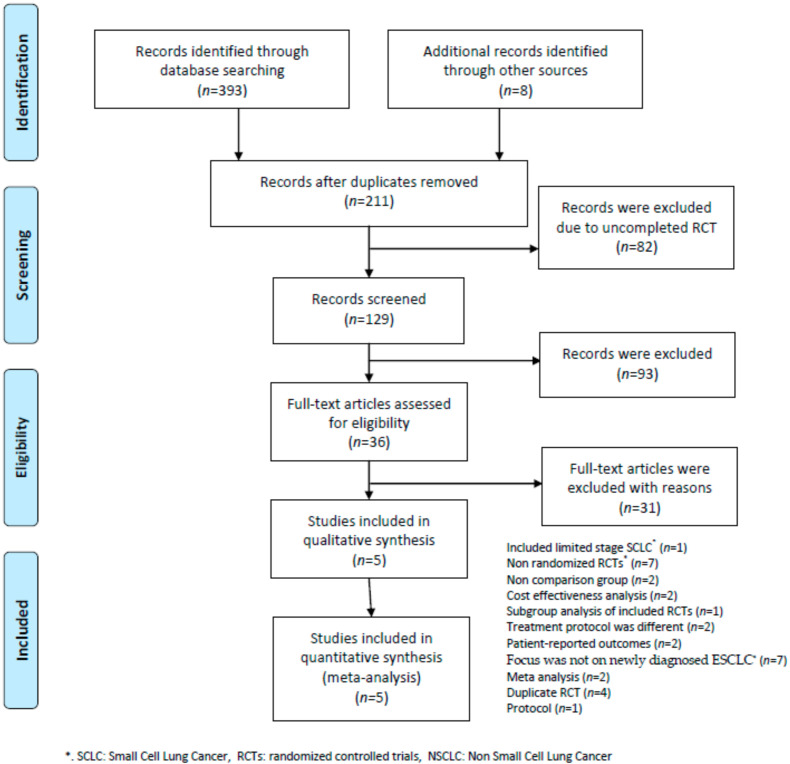
PRIMSA flow diagram.

**Figure 2 cancers-12-03629-f002:**
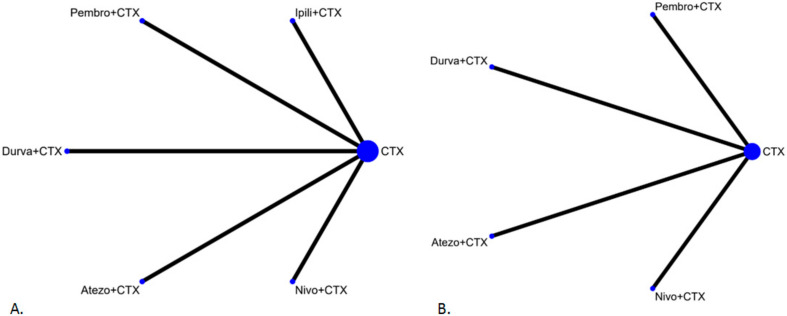
Network construction. (**A**) Network constructions for comparison in overall survival and grade 3–4 adverse events; (**B**) Network constructions for comparison in progression free survival and objective response ratio.

**Figure 3 cancers-12-03629-f003:**
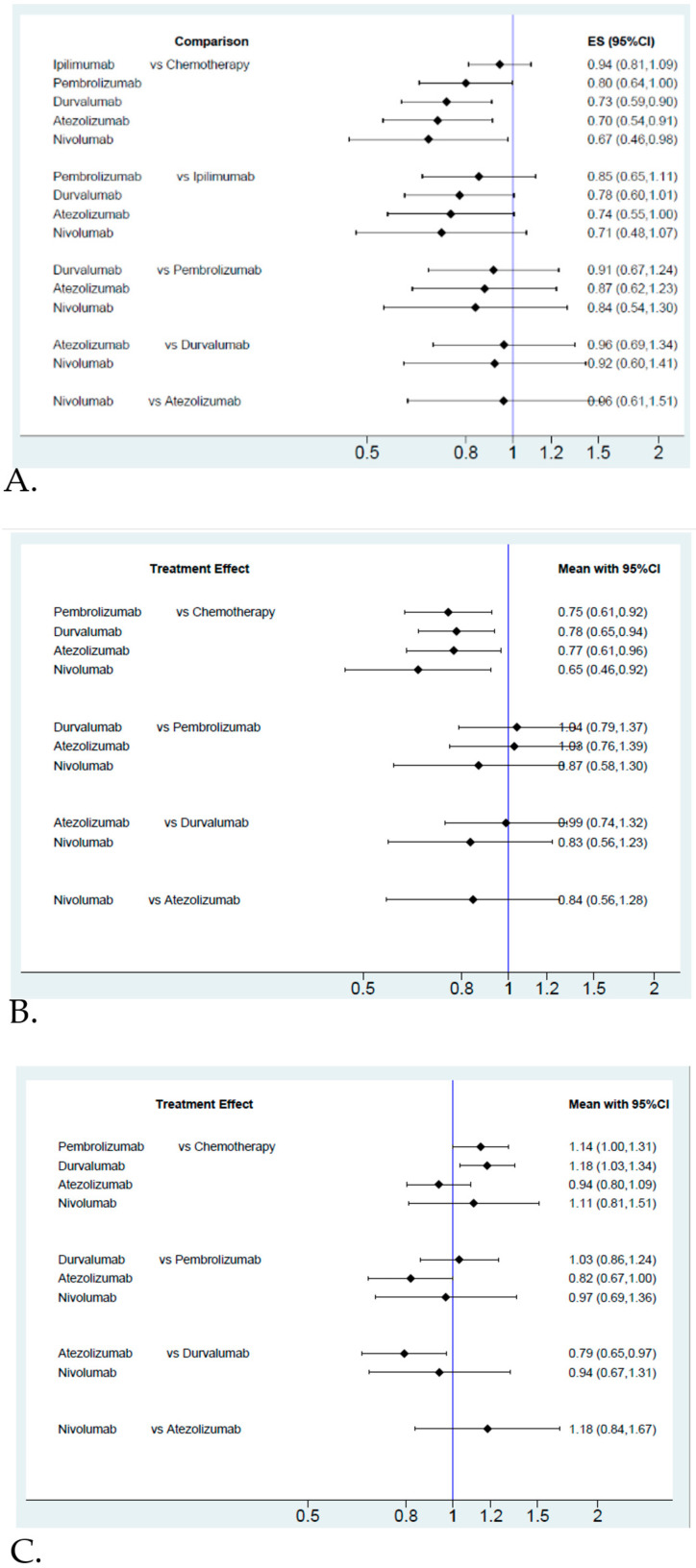
Summary of effect sizes from pairwise comparisons. (**A**) Hazard Ratio for overall survival; (**B**) Hazard Ratio for progression free survival; (**C**) Response Ratio for objective response rate; (**D**) Risk Ratio for grade 3–4 adverse events.

**Figure 4 cancers-12-03629-f004:**
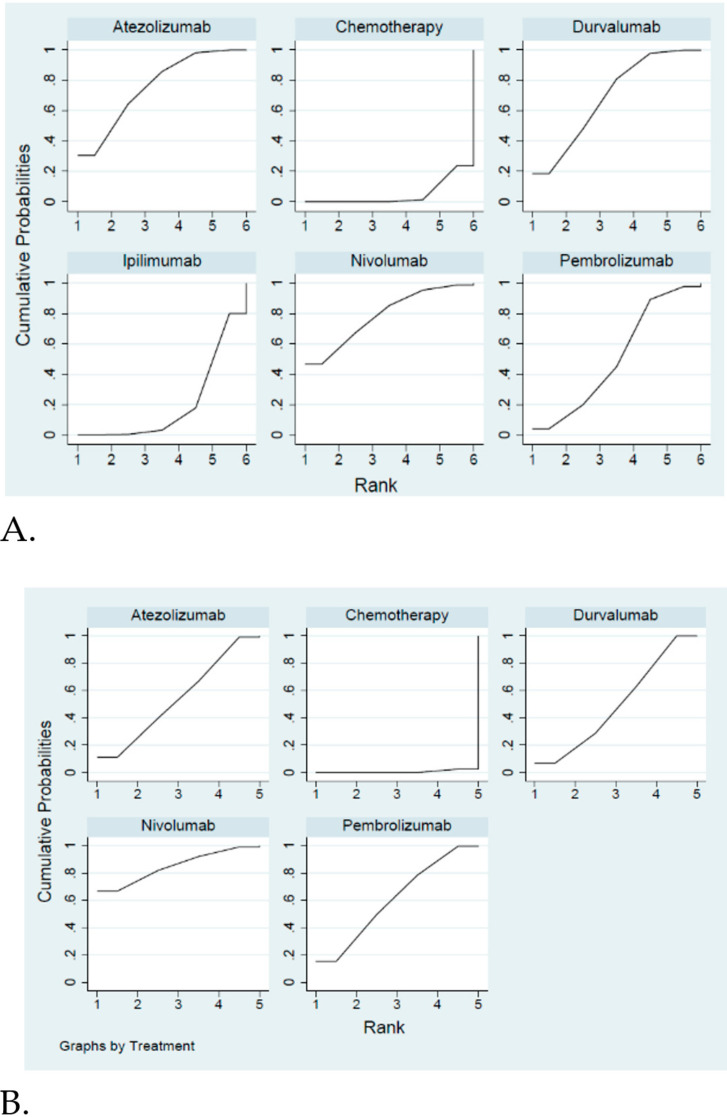
Cumulative ranking probability for different treatments. (**A**) Overall survival; (**B**) Progression free survival; (**C**) Objective response rate; (**D**) Grade 3–4 adverse events.

**Figure 5 cancers-12-03629-f005:**
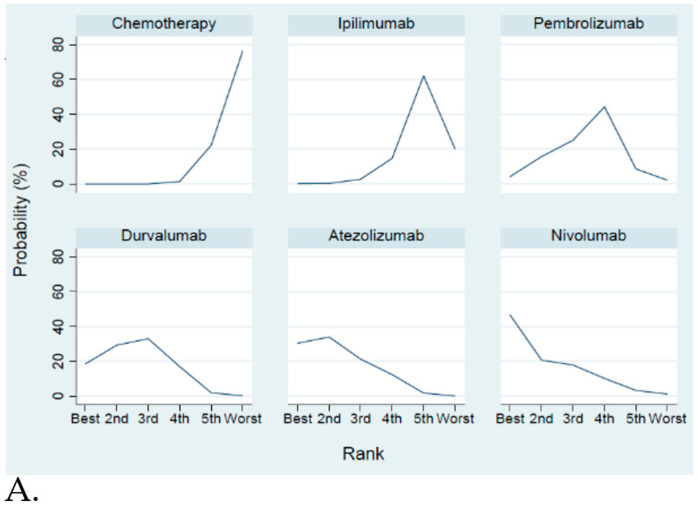
Probability to be the best treatment for different treatments. (**A**) Overall survival; (**B**) Progression free survival; (**C**) Objective response rate; (**D**) Grade 3–4 adverse events.

**Table 1 cancers-12-03629-t001:** Characteristics of the included studies.

Trial Name	ICI-Based Treatment	Year	Phase	Age ≥ 65 (%)	Males (%)	Brain or CNS * Meta (%)	Experimental Arm 1 (Number)	Experimental Arm 2 (Number)	Control Arm (Number)	Criteria for Progression or Response	Criteria for AE *
PD-1 inhibitors											
Keynote-604	pembrolizumab	2020	3	52.3%	64.9%	12.14%	(*n* = 228)		(*n* = 225)	RECIST *	CTCAE *
[18]							Induction: 4 × 21 days cycle		EP alone	version 1.1	
							Pembrolizumab + EP *		4 × 21 days cycle		
							maintenance *				
							pembrolizumab every 3 weeks				
EA5161	nivolumab	2020	2	unknown	44.4%	11.25%	(*n* = 80)		(*n* = 80)	RECIST	CTCAE
[14]				median			induction: 4 × 21 days cycle		EP alone	version 1.1	
				Age = 65			Nivolumab + EP		4 × 21 days cycle		
							maintenance *				
							nivolumab every 2 weeks				
PD-L1 inhibitors											
IMpower133	atezolizumab	2018	3	46.2%	64.8%	8.68%	(*n* = 201)		(*n* = 202)	RECIST	CTCAE
[13]							induction: 4 × 21 days cycle		EP alone	version 1.1	
							atezolizumab + EP		4×21 days cycle		
							maintenance *				
							atezolizumab every 3 weeks				
CASPIAN	durvalumab	2019	3	39.7%	69.6%	10.24%	(*n* = 268)	(*n* = 268)	(*n* = 269)	RECIST	CTCAE
[15]							induction: 4 × 21 days cycle	induction: 4 × 21 days cycle	EP alone	version 1.1	
							durvalumab + EP	durvalumab + EP + tremelimumab	4 × 21 days cycle		
							Maintenance *	Maintenance *			
							durvalumab every 4 weeks	durvalumab every 4 weeks			
CTLA4 inhibitors											
CA184-041	ipilimumab	2013	2	23.1%	78.4%	unknown	(*n* = 42)	(*n* = 42)	(*n* = 45)	1. mWHO *	CTCAE
[16]							phase regman 6 × 21 days cycle	concurrent regman 6 × 21 days cycle	paclitaxel + carboplatin 4 × 21 days cycle		
							paclitaxel + carboplatin at cycle 1–2	ipilimumab +paclitaxel +		2. irRC *	
							ipilimumab +paclitaxel + carboplatin at cycle 3–6	carboplatin at cycle 1–4			
								paclitaxel + carboplatin at cycle 5–6			
							maintenance *	maintenance *			
							ipilimumab every 12 weeks	ipilimumab every 12 weeks			
CA184-156	ipilimumab	2016	3	39.6%	56.9%	10.48%	(*n* = 478)		(*n* = 476)	1. mWHO	CTCAE
[17]							phase regman: 6 × 21 days cycle		EP alone 4 × 21 days cycle		
							EP at cycle 1–2			2. irRC	
							ipilimumab + EP at cycle 3–6				
							maintenance *				
							Ipilimumab every 12 weeks				

* CNS: central nervous system, EP: etoposide and platinum, AE: adverse event, RECIST: Response evaluation criteria in solid tumors version, CTCAE: Common Terminology Criteria for Adverse Events, mWHO: modified World Health Organization criteria, irRC: immune-related response criteria.

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
