# Peer review of "Systematic Review and Network Meta-Analysis of Immune Checkpoint Inhibitors in Combination with Chemotherapy as a First-Line Therapy for Extensive-Stage Small Cell Carcinoma"

_cancers, 2020, doi:10.3390/cancers12123629_

Round 1
Reviewer 1 Report
I would like to congratulate the authors for their valuable work. I believe this systematic review can be publishable if the authors are willing to take into consideration some of the suggestions/comments below
Abstract:
- Can the authors define a primary and secondary objective/endpoint for their work
Manuscript
Introduction
- The authors may want to add briefly the role of radiation in ES-SCLC (chest and brain..)
- The authors can mention the survival benefit seen in the RCTs conducted with IO/chemo in the first line setting in ES-SCLC so they can describe further why they decided to conduct their analysis
Results
- PRISMA chat: the numbers do not add up (36 to 19 to 5...)
- The authors may want to elaborate further about safety and efficacy (hazard ratio...) not only in the appendices (3, and 4) before report their network analysis. I also suggested report on the subgroup analyses reported
- Would the authors be able to report on specific grade 3/4 toxicities as well?
Discussion
- The references for traditional meta-analyses are not matching (16, 22, 24-27)
- The authors may want to discuss further the PFS and OS benefit with IO/chemo. The clinical relevant and the effect size are important to discuss
- I also suggest discussing why the survival benefit was not as striking as for other sites (NSCLC...) or at least what the authors think
- Similarly, I suggest discussing the discrepancy in ORR benefit across IO agents. What do the authors think about this?
- Other limitations to consider is the fact that some trials are reported as abstracts only so far
- The authors needs to discuss future directions. What are their recommendations based on the results they presented? and why
- Why the results may be diluted? The authors need to discuss some possible hypotheses or explanations
Author Response
Thank you so much for your affirmation, I greatly appreciate you taking the time to write such a detailed response. Our replies are presented point by point below. Again, thank you for giving us the opportunity to strengthen our manuscript with your valuable comments.

Reviewer 2 Report
In this meta-analysis using the published data (the individual patient data was not collected), the authors evaluated the efficacies (OS, PFS, and ORR) and adverse effects of adding immune checkpoint inhibitor (ICI) to cytotoxic chemotherapy (etoposide plus platinum: EP). The authors reported that EP plus nivolumab, atezolizumab or durvalumab had significant benefits compared with EP alone in terms of OS, and the probability of nivolumab being ranked first. In terms of PFS, all ICIs showed a longer PFS compared with EP alone, and nivolumab was again ranked first. The reviewer thinks the topic of this study is clinically important, however, the reviewer also raises some major and minor comments as summarized below.
- In Figure 1, 36 papers were assessed for full-text review and 19 articles were excluded with reasons. However, only 5 studies were left in the next box (studies included in qualitative synthesis). How about the other 12 papers??
- The reasons for excluding 19 papers above should be summarized in a Supplementary material.
- In Figure 3, the 95% CIs overlap between ICIs, therefore, the reviewer wonders if it is adequate that the authors described the manuscript suggesting that nivolmab was the most effective ICI together with EP for ED-SCLCs.
- The patient backgrounds (performance status, the presence of brain metastases, age, sex, PD-L1 expression status, etc) should be incorporated when the authors compare the efficacies between ICIs.
- The reviewer also suggests adding a similar comparison for the second line ICI for SCLC patients after chemotherapy failure. Is nivolmab the best in the second line setting, too?
Author Response
Thank you for giving us the opportunity to strengthen our manuscript with your valuable comments. We have tried our best to incorporate your feedback and hope that these revisions have improved the manuscript so that it is now acceptable for publication.

Round 2
Reviewer 2 Report
The reviewer thinks the authors responded well to the comments by the reviewers.